# Cyanocobalamin prevents cardiomyopathy in type 1 diabetes by modulating oxidative stress and DNMT-SOCS1/3-IGF-1 signaling

Masao Kakoki [1✉], Purushotham V. Ramanathan[1], John R. Hagaman[1], Ruriko Grant[1], Jennifer C. Wilder[1], Joan M. Taylor[1], J. Charles Jennette[1], Oliver Smithies[1] & Nobuyo Maeda-Smithies[1]

Patients with long-standing diabetes have a high risk for cardiac complications that is exacerbated by increased reactive oxygen species (ROS) production. We found that feeding cyanocobalamin (B12), a scavenger of superoxide, not only prevented but reversed signs of cardiomyopathy in type 1 diabetic $Elmo1^{H/H}$ $Ins2^{Akita/+}$ mice. ROS reductions in plasma and hearts were comparable to those in mice treated with other antioxidants, N-acetyl-L-cysteine or tempol, but B12 produced better cardioprotective effects. Diabetes markedly decreased plasma insulin-like growth factor (IGF)-1 levels, while B12, but not N-acetyl-L-cysteine nor tempol, restored them. B12 activated hepatic IGF-1 production via normalization of S-adenosylmethionine levels, DNA methyltransferase (DNMT)-1/3a/3b mRNA, and DNA methylation of promoters for suppressor of cytokine signaling (SOCS)-1/3. Reductions of cardiac IGF-1 mRNA and phosphorylated IGF-1 receptors were also restored. Thus, B12 is a promising option for preventing diabetic cardiomyopathy via ROS reduction and IGF-1 retrieval through DNMT-SOCS1/3 signaling.

[1] Department of Pathology and Laboratory Medicine, The University of North Carolina at Chapel Hill, Chapel Hill, NC, USA. ✉email: mkakoki@med.unc.edu

Diabetic cardiomyopathy is a clinical entity of global cardiac dysfunction, which is associated with diabetes mellitus but not caused by hypertensive, coronary or valvular heart disease[1]. Despite the increased heart failure risks in both type 1 and type 2 diabetic patients, genetic/molecular analyses in human patient populations have been difficult in part because of the overwhelmingly high frequency of other concomitant risk factors for cardiovascular diseases. In contrast, various animal models of diabetes have confirmed the development of diabetes-induced cardiomyopathy with diastolic dysfunction[2,3], providing platforms for the dissection of genetic and environmental factors involving in the pathological processes.

Engulfment and cell motility protein 1 (ELMO1) physically interacts with the dedicator of cytokinesis 180 (DOCK180), a member of guanine nucleotide exchange factors (GEFs), to activate small GTPase Ras-related C3 botulinum toxin substrate (Rac)[4,5]. The activated Rac is an essential component of NADPH oxidase (NOX) 1 and 2[6,7], which generate superoxide in many cell types. Polymorphisms in the $ELMO1$ gene are associated with diabetic nephropathy in humans[8], and $Elmo1$ expression levels are directly correlated with severity of nephropathy in diabetic mice[9]. We further reported that mice with twice the normal expression of $Elmo1$ ($Elmo1^{H/H}$) in Akita diabetic background ($Ins2^{Akita/+}$) develop severe cardiac dysfunction by about 16 weeks of age[10]. Enhanced oxidative stress, severely impaired cardiac contractile and diastolic function, dissociation of intercalated discs, and mitochondrial fragmentation were among the prominent characteristics, although these features are also present to a lesser extent in Akita mice with normal expression of $Elmo1$[10]. In contrast, Akita mice with 30% normal expression of $Elmo1$ ($Elmo1^{L/L}$) were completely protected[10]. Thus, ELMO1 is a critical factor for reactive oxygen species (ROS) production via NOXs, which in turn trigger cellular signaling cascades towards tissue damage in diabetes mellitus. As the early development of cardiac changes mimicking those of humans is readily detectable, and because non-diabetic $Elmo1^{H/H}$ $Ins2^{+/+}$ mice do not develop the cardiac disease[10], $Elmo1^{H/H}$ $Ins2^{Akita/+}$ mice represent an excellent experimental model for complications of type 1 diabetes.

Since enhanced ROS production is the major factor of diabetic cardiomyopathy in the $Elmo1^{H/H}$ $Ins2^{Akita/+}$ mice, we have chosen in the current study to evaluate the effects and mechanisms of cobalamins (B12) on this devastating diabetic complication. B12, an essential nutrient for our life and generally regarded as safe, is also an efficient SOD mimetic[11,12]. The precursor of most ROS in biological systems is the superoxide anion free radical, $O_2^-$, normally kept in check enzymatically by superoxide dismutases (SODs), which catalyze its conversion into molecular oxygen ($O_2$) and hydrogen peroxide ($H_2O_2$). Considering its superoxide scavenging property, protective effects of B12 in ROS-induced cell injuries is promising. Yet the documentation of its oral therapeutic use has been scarce, in part because its potential for inactivating ROS is normally limited because absorption of B12 from the diet is controlled by the small amount of intrinsic factor secreted by the gastric mucosa. However, dietary B12 at high doses enables absorption of the vitamin by an intrinsic-factor-independent but low-efficiency pathway that overcomes this limitation[13,14]. Indeed, we have recently reported that high-dose oral supplementation of B12 in drinking water moderated renal superoxide production and post-ischemia/reperfusion injury in mice[15].

In this paper, we demonstrate that oral administration of high-dose cyanocobalamin (B12) markedly reduced circulating/tissue ROS and prevented as well as reversed structural and functional changes in $Elmo1^{H/H}$ $Ins2^{Akita/+}$ mice. The cardioprotective effects of B12 on the diabetic hearts were greater than those of N-acetyl-L-cysteine (NAC), an antioxidant amino acid, and those of

4-hydroxyl-tetramethylpiperidin-oxyl (tempol), a superoxide dismutase (SOD) mimetic, although all of B12, NAC and tempol comparably suppressed the markers of ROS in $Elmo1^{H/H}$ $Ins2^{Akita/+}$ mice. This led us to further uncover that oral high-dose B12, but not NAC nor tempol, restored hepatic insulin-like growth factor-1 (IGF-1) production and exerted beneficial effects on the hearts of $Elmo1^{H/H}$ $Ins2^{Akita/+}$ mice. B12 is the essential cofactor for methionine synthase in the folate-methionine cycle, which affects S-adenosylmethionine (SAMe) production, a methyl donor for DNA methyltransferase (DNMT). Enhanced DNA methylation of the genes for suppressor of cytokine signaling (SOCS) 1 and 3 in turn increased hepatic production of IGF-1. Our data thus suggest that high-dose B12 is a promising option to prevent diabetic nephropathy by both mitigating oxidative stress and by restoring the SAMe-DNMT-SOCS-IGF-1 signaling cascade.

## Results

**Oral administration of high doses of B12 prevents diabetic cardiomyopathy in $Elmo1^{H/H}$ $Ins2^{Akita/+}$ mice.** We first determined the dose effects of orally administered B12 on the plasma and cardiac levels of B12 and on the cardiac function in the diabetic $Elmo1^{H/H}$ $Ins2^{Akita/+}$ mice. Plasma and cardiac levels of B12 were lower in diabetic $Elmo1^{H/H}$ $Ins2^{Akita/+}$ mice than in non-diabetic $Elmo1^{H/H}$ $Ins2^{+/+}$ mice (down to ~50% and 30% respectively, Suppl. Figs. 1a, b). We provided 8-week-old $Elmo1^{H/H}$ $Ins2^{Akita/+}$ mice with B12 at 1, 10, and 100 mg/kg body weight/day (kg bw/day) in their drinking water for 8 weeks. At the end of 8 weeks, plasma and cardiac levels of B12 were incrementally increased and at a dose of 10 mg/kg bw/day the levels were normalized back to the levels of non-diabetic mice (shown by the dotted lines, Fig. 1a, b). Supplementation at 100 mg/kg bw/day, in contrast, average levels were higher than in non-diabetic mice at ~130%.

Administration of B12 for 8 weeks did not induce differences in BW, systolic arterial pressure, heart weight normalized by tibia length, plasma total cholesterol levels, or hematocrit (Supplemental Data 1). Diabetic mice also had higher plasma levels of glucose and triglycerides than non-diabetic mice, and the treatment with B12 did not change those levels (Supplemental Data 1).

Echocardiographic parameters, the ejection fraction of the left ventricle (LVEF; Fig. 1c), the left ventricular posterior wall thickness in diastole (LVPWd; Fig. 1d), and the internal diameter of the left ventricle in diastole (LVIDd; Fig. 1e), indicate dilatation of hearts in the $Elmo1^{H/H}$ $Ins2^{Akita/+}$ mice compared to those in non-diabetic wild type mice (illustrated by dotted lines in the figures). They were improved in the $Elmo1^{H/H}$ $Ins2^{Akita/+}$ mice given B12 at the dose of 1, 10 or 100 mg/kg/day. The parameters indicating diastolic function including E-wave deceleration rate (EWDR), isovolumic relaxation time (IVRT), and E' were also normalized by B12 (Fig. 1f–h). Taken together, our data demonstrate that high-dose oral supplements of B12 prevent development of diabetic cardiomyopathy in the $Elmo1^{H/H}$ $Ins2^{Akita/+}$ mice. As B12 at the dose of 10 mg/kg bw/day normalized the plasma and tissue levels and improved the LVEF, LVPWd, LVIDd, EWDR, IVRT, and E', we used this dose in the following experiments.

**A high-dose B12 prevents and restores cardiac functions in the $Elmo1^{H/H}$ $Ins2^{Akita/+}$ mice already showing decline.** Typically, $Ins2^{Akita/+}$ mice become maximally hyperglycemic by 8 weeks of age. Our experiments above and our earlier work[10] demonstrated that $Elmo1^{H/H}$ $Ins2^{Akita/+}$ mice develop severe cardiomyopathy by age 16 weeks. To examine whether B12 supplementation can

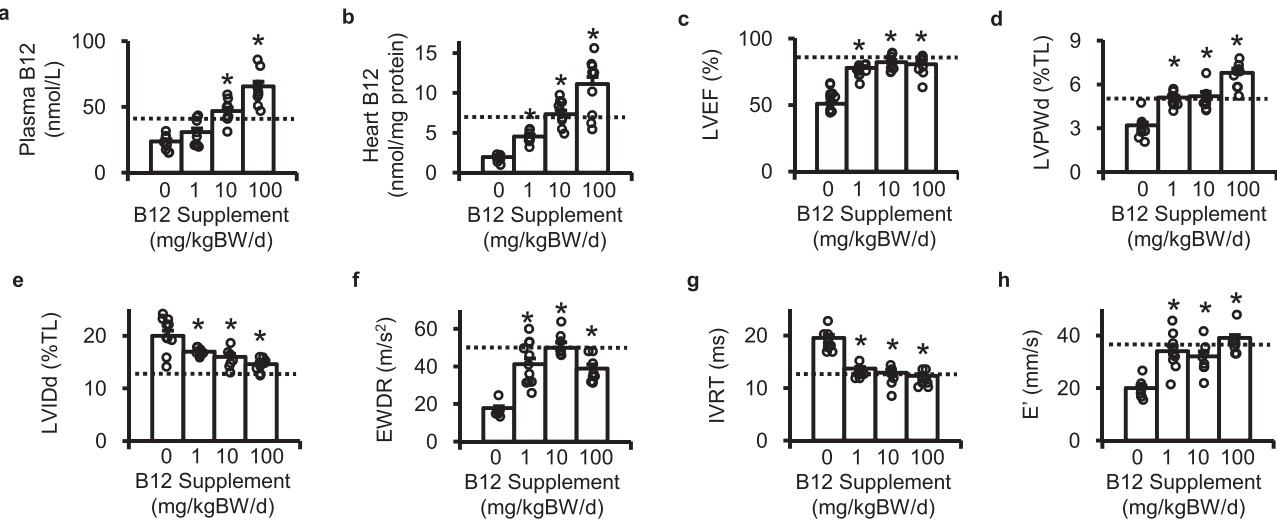

**Fig. 1 Oral high-dose cyanocobalamin (B12) prevents diabetic cardiomyopathy.** B12 was supplied to the *Elmo1^{H/H} Ins2^{Akita/+}* mice in drinking water to achieve estimated daily intake of 0, 1, 10 and 100 mg/kg body weight (BW). **a** Plasma levels of B12. **b** Cardiac levels of B12. **c** Ejection fraction of the left ventricles (LVEF). **d** Left ventricular posterior wall thickness in diastole (LVPWd). **e** Internal diameter of the left ventricle in diastole (LVIDd). **f** E-wave deceleration rate (EWDR) of the mitral flow. **g** Isovolumic relaxation time (IVRT) of the left ventricle **h** Early tissue Doppler velocity (E'). Data are expressed as means ± standard errors. Nine to ten mice were in each group. *$P < 0.05$ vs. mice with non-supplemented drinking water by Tukey–Kramer Honestly Significant Differences test after one-way ANOVA. LVPWd and LVIDd were normalized by tibia length (TL). Dotted lines in the figures indicate average levels in non-diabetic *Elmo1^{H/H} Ins2^{+/+}* mice.

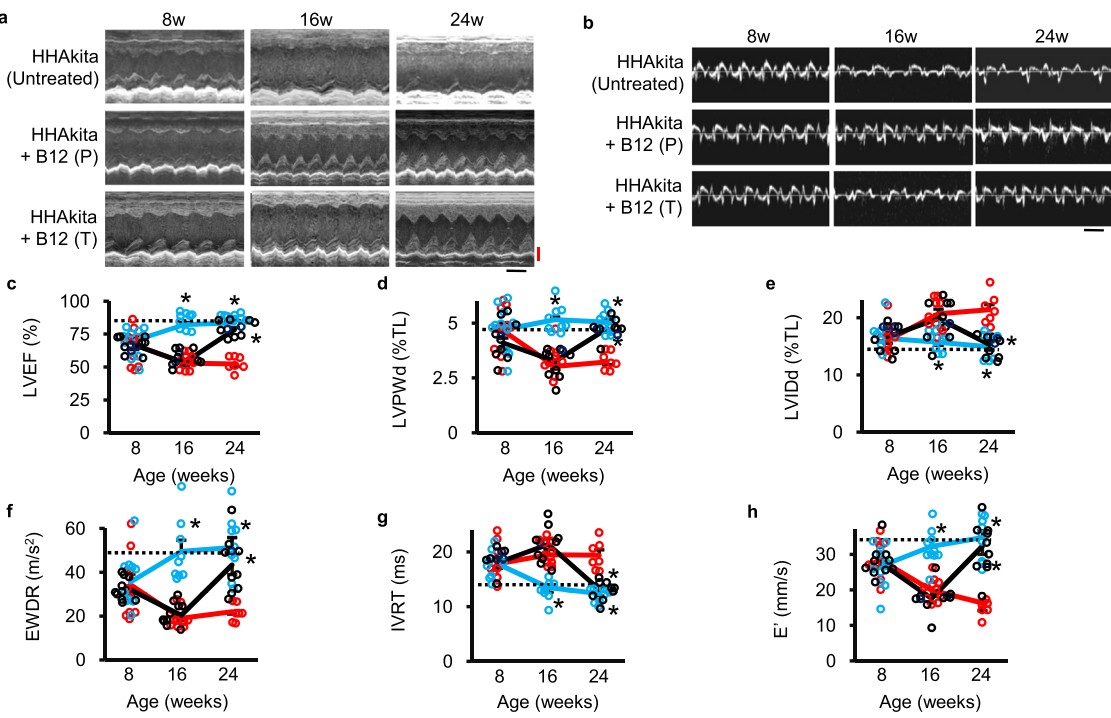

**Fig. 2 Prevention and reversal of diabetic cardiomyopathy in the Elmo1H/H Ins2Akita/+ (HHAkita) mice treated with B12 at 10 mg/kg bw/day.**
**a** Representative M-mode echocardiograms. Black bar, 0.1 s; Red bar, 0.1 mm. **b** Representative tissue Doppler imaging. Black bar, 0.1 s; Red bar, 50 mm/s. **c** LVEF, **d** LVPWd, **e** LVIDd, **f** EWDR, **g** IVRT and **h** E' in mice at age 8, 16, and 24 weeks. HHAkita mice were given 10 mg/kg bw/day of B12 in the drinking water starting at 8 weeks of age (preventive protocol, B12P; blue lines; $N = 9$) or starting at 16 weeks of age (therapeutic protocol, B12T; black lines; $N = 10$). Data are expressed as means ± standard errors. The numbers of untreated mice (red lines) were 10 at age 8 and 16 weeks and 7 at age 24 weeks).
*$P < 0.05$ vs. mice with non-supplemented drinking water by Tukey–Kramer Honestly Significant Differences test. Dotted lines in the figures indicate average levels in non-diabetic *Elmo1^{H/H} Ins2^{+/+}* mice.

improve already established cardiomyopathy, we randomly assigned 8-week-old *Elmo1^{H/H} Ins2^{Akita/+}* to three groups; (1) non-supplemented, (2) B12-supplemented from 8 weeks of age, and (3) B12-supplemented from 16 weeks of age. Heart

functions were monitored with echocardiography at 8, 16, and 24 weeks of age. As shown in Fig. 2a, c, LVEF in non-supplemented mice in group 1 (red line) gradually declined throughout, and 3 mice out of 11 mice died between age 16 and

24 weeks. In contrast, cardiac function of mice given B12 from the onset of diabetes at 8 weeks (group 2, blue line) improved and sustained normal levels of LVEF. In the group 3 mice that began to receive B12 at 16 weeks of age, when their heart function already showed a decline, an 8-week treatment clearly restored their heart function (black line). No mice in this group died. Thus, B12 prevents, as well as reverses diabetic cardiomyopathy. Likewise, the B12-treated $Elmo1^{H/H} Ins2^{Akita /+}$ mice had greater LVPWd (Fig. 2d) and less LVIDd (Fig. 2e) than the untreated $Elmo1^{H/H} Ins2^{Akita/+}$ mice in both group 2 (preventive) and group 3 (therapeutic). Other measures of diastolic function such as EWDR (Fig. 2f), IVRT (Fig. 2g), and E' (Fig. 2b, h) were also improved in the B12-treated mice.

The beneficial effect of B12 on diabetic cardiomyopathy is independent of $Elmo1$ genotype. Echocardiograms revealed that the LVEF in the $Elmo1^{+/+} Ins2^{Akita/+}$ mice ($76 \pm 1\%$, $n = 9$) was reduced was compared with wild type ($Elmo1^{+/+} Ins2^{+/+}$) mice ($85 \pm 1\%$, $n = 9$) at the age of 16 weeks (Suppl. Fig. 2a), although the degrees of reduction were small compared to those in the $Elmo1^{H/H} Ins2^{Akita/+}$ mice ($51 \pm 2\%$, $n = 10$; Fig. 1c). B12 treatment normalized the LVEF. Likewise, the EWDR of the mitral flow and the E' were less and the IVRT was higher in the untreated $Elmo1^{+/+}Ins2^{Akita /+}$ mice than in the $Elmo1^{+/+} Ins2^{+/+}$ mice. B12 treatment restored diabetic effects in these parameters in the $Elmo1^{+/+} Ins2^{Akita /+}$ mice (Suppl. Fig. 2b–f).

These data demonstrate that oral administration of a high-dose B12 can both prevent and reverse early signs of cardiac dysfunction in Akita diabetic mice regardless of the levels of $Elmo1$ expression.

**Cardioprotective effects of orally administered B12 in the $Elmo1^{H/H}Ins2^{Akita /+}$ mice exceeds those of NAC, and tempol**. Previous works have shown the cardioprotective effects of anti-oxidants in diabetic rodents[16,17]. To compare the preventive effects of oral B12 at 10 mg/kg bw/day with those of other anti-oxidants, NAC (1000 mg/kg bw/day) and tempol (200 mg/kg bw/day) were administered through drinking water, starting at 8 weeks of age. Cardiac function was evaluated after 8 weeks of treatment and other factors were characterized after 16 weeks of treatment (24 weeks of age). Treatments with all these anti-oxidants comparably reduced ROS parameters including cardiac immunoreactivity of 4-hydroxy-2-nonenal (4-HNE), cardiac $H_2O_2$ release, and plasma levels of thiobarbiturate-reactive substances (TBARSs), and increased the ratio of reduced glutathione (GSH) to oxidized glutathione (GSSG) in the $Elmo1^{H/H} Ins2^{Akita /+}$ mice (Fig. 3a, b and Suppl. Fig. 3a, b). B12 and other ROS scavengers, NAC and tempol, improved the pressure–volume (PV) loop and echocardiographic parameters for diastolic function (Fig. 3c, d and Suppl. Fig. 3c–h) without changing blood pressures or plasma levels of glucose, cholesterol, insulin or triglycerides (Suppl. Table 1).

Despite the comparative antioxidative effects by NAC, tempol and B12, the cardioprotective effect by B12 was more potent than NAC and tempol. Thus, we noted that the magnitude of protections judged by PV loop and echocardiographic parameters by NAC and tempol was less than that achieved by B12 (Fig. 3c, d and Suppl. Fig. 3c–h). Heidenhain's AZAN trichrome staining revealed that fibrotic areas of the heart were greater in diabetic $Elmo1^{H/H} Ins2^{Akita /+}$ mice compared to non-diabetic $Elmo1^{H/H} Ins2^{+/+}$mice (Fig. 3e, g). Histological staining by wheat germ agglutinin lectin of the heart revealed that the cross-sectional areas of cardiomyocytes were smaller in untreated diabetic $Elmo1^{H/H} Ins2^{Akita /+}$ mice compared to non-diabetic $Elmo1^{H/H} Ins2^{+/+}$ mice (Fig. 3f, h). B12 prevented the size reduction almost completely in the $Elmo1^{H/H} Ins2^{Akita/+}$ mice, while prevention by

NAC and tempol were partial. None of these antioxidants affected the cardiomyocyte cross-sectional areas in non-diabetic $Elmo1^{H/H} Ins2^{+/+}$ mice (Fig. 3e–h).

We previously reported that the frequency of intercalated disc dissociation and the magnitude of mitochondrial fragmentation are increased by 16 weeks in the $Elmo1^{H/H} Ins2^{Akita/+}$ mice[10]. Transmission electron microscopy (TEM) revealed that the frequency of intercalated disc dissociation was less in B12-treated $Elmo1^{H/H} Ins2^{Akita/+}$ mice than in untreated $Elmo1^{H/H} Ins2^{Akita/+}$ mice (Fig. 4a, b). TEM also revealed that the cross-sectional sizes of mitochondria were larger and mitochondrial number was less in B12-treated $Elmo1^{H/H} Ins2^{Akita/+}$ mice than in untreated mice (Fig. 4c and Suppl. Fig. 3i). Other antioxidants also showed protection of both intercalated disc damage and mitochondria of cardiomyocytes, but the protection was less than those by B12. The volume density of mitochondria was comparable among untreated and antioxidant-treated $Elmo1^{H/H} Ins2^{Akita/+}$ mice (Suppl. Fig. 3j). B12 treatment completely restored cardiac citrate synthase activity and complex I, IV, and V activities that were markedly reduced in untreated diabetic mice (Fig. 4d–g). Both tempol and NAC improved all these parameters, but the degree of improvement in cardiac mitochondrial function was not as great as those in B12-treated animals.

**Oral supplementation with a high-dose B12, but not with NAC or tempol, restores hepatic insulin-like growth factor (IGF)-1 production in the ELMO1 hypermorphic mice**. We next investigated gene expression profiles to approach the mechanism whereby B12 provides more cardioprotective effects than those by NAC and tempol. Consistent with reduced cardiac functions in $Elmo1^{H/H} Ins2^{Akita/+}$ mice, diabetes increased cleaved caspase 3 protein levels (Fig. 5a and Suppl. Figs. 4a and 7a, b) and the mRNA levels of cardiomyopathy-associated genes, including the p53 gene ($Trp53$)[18], β-myosin heavy chain gene ($Myh7$)[19] (Fig. 5b, c), and transforming growth factor-β1 gene ($Tgfb1$)[20] (Suppl. Fig. 4b), while it decreased mRNA levels of α-myosin heavy chain ($Myh6$; Fig. 5d). These changes of the gene expression in the $Elmo1^{H/H} Ins2^{Akita/+}$ mice were partially reversed (about 50%) by NAC and tempol, and almost totally reversed by B12 (Fig. 5a, d). These data suggest that B12 exhibits additional protective properties to reduction of ROS in our diabetic model.

Type 1 diabetes mellitus is associated with decreased circulating levels of IGF-1[21,22]. Furthermore, the transgenic expression of IGF-1 and the intraperitoneal injection of IGF-1 have been shown to protect against cardiac dysfunction in mice with type 1 diabetes[18]. Accordingly, IGF-1 signaling is a candidate mechanism that B12 mediates cardioprotection. We observed that plasma concentrations of IGF-1 and both hepatic and cardiac levels of IGF-1 mRNA were markedly lower in untreated diabetic mice than non-diabetic mice (Fig. 5e–g). Oral administration of B12 restored plasma IGF-1 and hepatic IGF-1 mRNA levels in the diabetic $Elmo1^{H/H} Ins2^{Akita/+}$ mice raising to the levels in non-diabetic $Elmo1^{H/H} Ins2^{+/+}$ mice. In contrast, NAC and tempol did not have any effects on these levels (Fig. 5e–g). Additionally, cardiac protein levels of phosphorylated IGF-1 receptor β (pIGF1Rβ), which reflects intracellular IGF-1 signaling, were markedly lower in untreated diabetic mice than in non-diabetic mice (Fig. 5h, and Suppl. Figs. 4b and 7c, d). Oral administration of B12 restored cardiac pIGF1Rβ levels in the diabetic $Elmo1^{H/H} Ins2^{Akita/+}$ mice to the levels in the untreated $Elmo1^{H/H} Ins2^{+/+}$ mice, but neither NAC nor tempol did (Fig. 5h and Suppl. Figs. 4b and 7c, d). In all cases, the changes of IGF-1 related parameters by B12 achieved in the diabetic mice did not exceed the levels seen in non-diabetic mice. We further note that B12 treatment, but not NAC nor tempol, resulted in a small

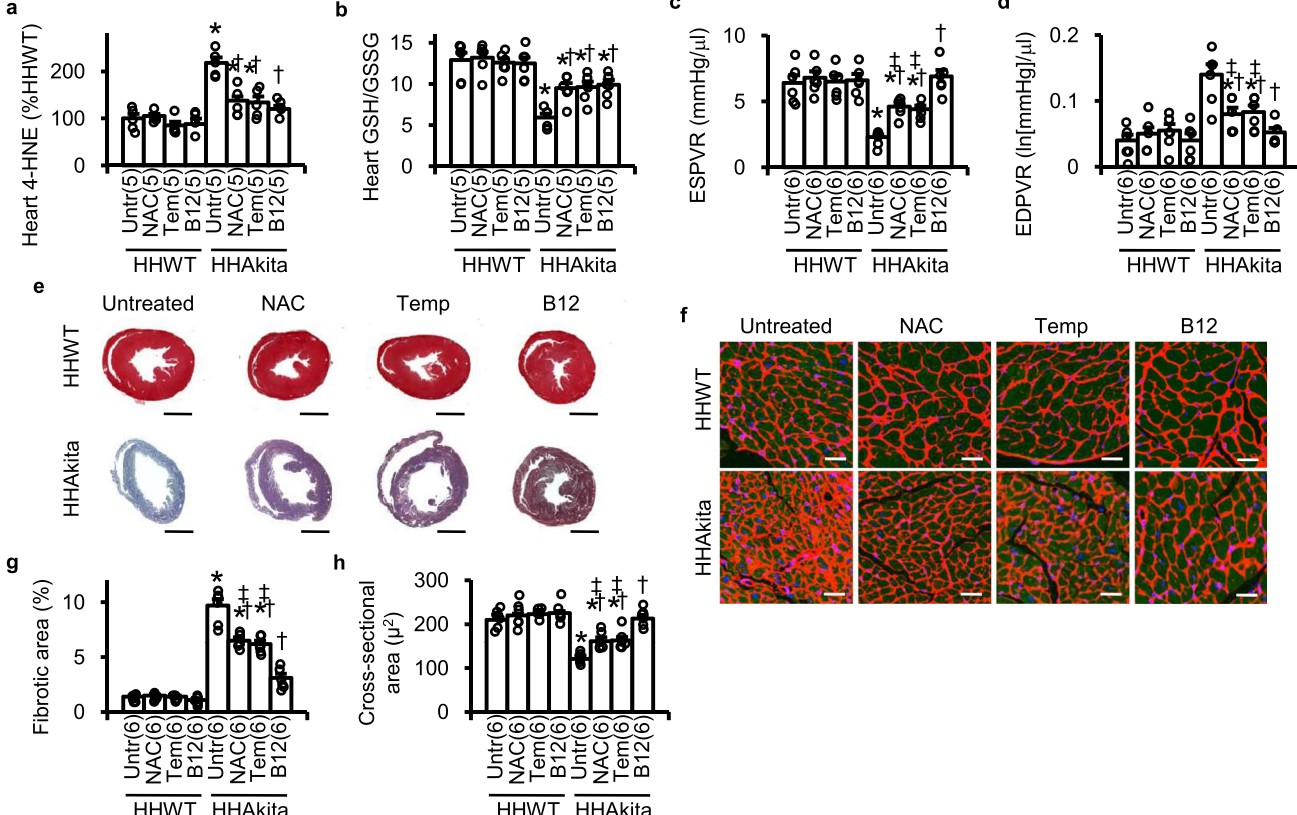

**Fig. 3 Antioxidative and cardioprotective effects of B12 (10 mg/kg/day) compared to those of N-acetyl-ʟ-cysteine (NAC; 1000 mg/kg/day), and tempol (Tem; 200 mg/kg/day).** Mice having high expression levels of ELMO1 without and with Akita diabetetogenic mutation (HHWT and HHAkita, respectively) were administered with NAC, Tem and B12 in their drinking water starting at 8 weeks of age for 16 weeks. **a** Cardiac levels of 4-hydroxy-2-nonenal (4-HNE). **b** The ratio of reduced glutathione (GSH) to oxidized glutathione (GSSG) in the heart. **c** End-systolic pressure–volume repationship (ESPVR). **d** End-diastolic pressure–volume repationship (ESPVR). **e** AZAN trichrome staining of the heart. Scale bar = 1 mm. **f** Representative photomicrographs of cross-sections of cardiomyocytes stained with wheat germ agglutinin (red), anti-α-actinin antibody (green) and 4′,6-diamidino-2-phenylindole (DAPI; blue). Scale bar = 10 μm. **g** Proportion of fibrotic areas. **h** Cross-sectional areas of cardiomyocytes. Data are expressed as means ± standard errors. Numbers in the parentheses indicate the number of animals studied. *$P < 0.05$ vs. untreated (Untr) HHWT mice by Tukey–Kramer Honestly Significant Differences test after one-way ANOVA; †$P < 0.05$ vs. untreated HHAkita mice. ‡$P < 0.05$ vs. B12-treated HHAkita mice.

increase in plasma IGF-1, hepatic and cardiac IGF-1 mRNA and cardiac pIGF1Rβ levels also in non-diabetic mice. These increases, however, did not result in any noticeable or measurable phenotypic changes in the B12-treated mice.

Although growth hormone (GH) stimulates the synthesis of IGF-1 in the liver[23], plasma GH levels were unchanged by the presence of diabetes or by B12 administration (Fig. 5i), indicating that the alteration of plasma IGF-1 levels is not due to the changes in the GH levels. In response to the binding of GH to GH receptors on hepatocytes, STAT5 becomes phosphorylated and translocated into the nucleus, which stimulates the transcription of IGF-1 in hepatocytes. In contrast to unaltered plasma GH levels, liver protein levels of phosphorylated STAT5 levels were decreased by diabetes. Again, B12 but not NAC or tempol restored pSTAT5 levels (Fig. 5j and Suppl. Figs. 4c and 7e, f).

**B12 modulates the DNA methylases (DNMTs)- suppressors of cytokine signaling (SOCS) 1/3 cascade to enhance IGF-1 signaling that mediates the cardioprotection.** B12 is an essential cofactor of methionine synthase in the folate-methionine cycle to generate methionine, which is a precursor of S-adenosylmethionine (SAMe), a methyl group donor for DNA and protein methylation. It has recently been reported that B12 supplementation induces genome-wide changes in DNA methylation[24]. In *Elmo1*$^{H/H}$ mice, the presence of Akita diabetes decreased hepatic levels of SAMe (Fig. 6a).

Oral administration of B12 restored hepatic levels of SAMe in the *Elmo1*$^{H/H}$ *Ins2*$^{Akita/+}$ mice. The same dose increased SAMe by about 40% in the *Elmo1*$^{H/H}$ *Ins2*$^{+/+}$ mice. Neither NAC nor tempol had such effects (Fig. 6a).

Previous studies demonstrated that the availability of SAMe changes in parallel to the expression of DNA methyltransferases (DNMTs)[25,26]. We additionally observed that mRNA levels of DNMT1, DNMT3a and DNMT3b were decreased by Akita diabetes but restored by B12 (Fig. 6b–d). DNMT1/3a/3b methylate DNA in the promoter of suppressor of cytokine signaling (SOCS) 1 and 3 genes and decrease their expression[27–32]. Concordant with these reports, we found that the promoters for SOCS1/3 are demethylated in Akita diabetes and methylated by B12 (Fig. 6e). In parallel, mRNA levels for SOCS1/3 were moderated by B12 in both *Elmo1*$^{H/H}$ *Ins2*$^{Akita/+}$ mice and *Elmo1*$^{H/H}$ *Ins2*$^{+/+}$ mice (Fig. 6f, g). As both SOCS1 and SOCS3 bind to Janus kinase (JAK) 2[33] and inhibit intracellular signaling of GH receptors, our findings suggest that DNA methylation in the SOCS1/3 promoters enhanced by B12 supplementation leads to restored production of IGF-1 and contributes to ROS-independent beneficial effects of B12 on diabetic cardiomyopathy.

Lastly, we studied whether the SAMe-DNMTs-SOCS1/3-IGF-1 cascade stimulated by B12 indeed mediates protection against diabetic cardiomyopathy in *Elmo1*$^{H/H}$ *Ins2*$^{Akita/+}$ mice. We

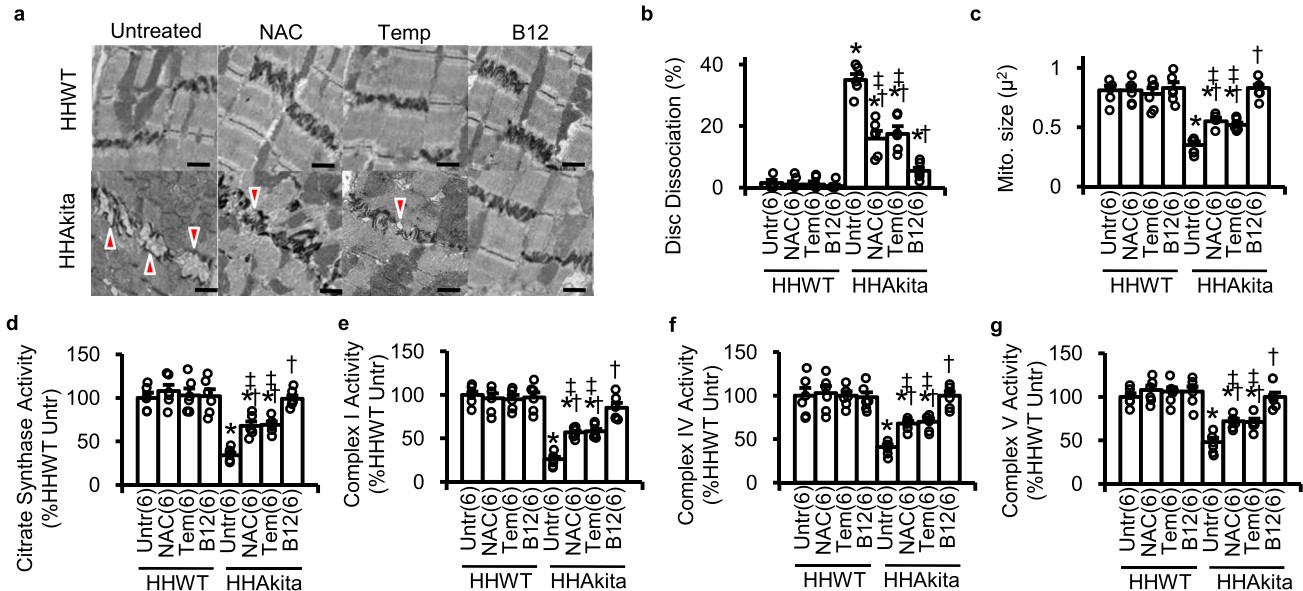

**Fig. 4 Effects of orally administered NAC, tempol (Tem), and B12 on the ultrastructure and mitochondrial function in the heart. a** Representative images of the intercalated disc in the heart of HHWT (*Elmo1^{H/H} Ins2^{+/+}*) and HHAkita (*Elmo1^{H/H} Ins2^{Akita/+}*) mice in the transmission electron microscopy. Arrowheads indicate the dissociation of intercalated discs. Scale bar = 1 μm. **b** Proportion of the dissociated intercalated discs. **c** Apparent size of mitochondria (mito.) were scored from at least 30 mitochondria in each animal. **d** Citrate synthase activity in the heart. **e** Mitochondrial complex I activity. **f** Mitochondrial complex IV activity. **g** Mitochondrial complex V activity. Data are expressed as means ± standard errors. $N = 6$ in each group. *$P < 0.05$ vs. untreated (Untr) HHWT mice by Tukey–Kramer Honestly Significant Differences test after one-way ANOVA; †$P < 0.05$ vs. untreated HHAkita mice; ‡$P < 0.05$ vs. B12-treated HHAkita mice.

observed that restoration in IGF-1 mRNA levels and methylation in the promoter of SOCS1/3 and moderation in mRNA levels of SOCS1/3 by B12 were fully recapitulated by SAMe, the methionine-derived methyl donor for DNMTs, and were almost completely inhibited by an inhibitor of DNMTs 5-aza-2′-deoxycytidine (Fig. 7a–d). A SOCS1/3 inhibitor pJAK2 (1001–1013) mimicked the restoration in IGF-1 mRNA levels by B12 (Fig. 7a).

Furthermore, oral co-administration of an inhibitor of IGF-1 signaling linsitinib or 5-aza-2′-deoxycytidine partially, but inhibited the beneficial effects of B12 on cardiac dysfunction (Fig. 7e, f and Suppl. Fig. 5a–d), cross-sectional area of cardiomyocytes, fibrotic area, proportion in dissociated intercalated disc, magnitude of the mitochondrial fragmentation, and mitochondrial function, including citrate synthase activity and activity of complex I, IV, and V (Suppl. Fig. 6a–j) in the heart of *Elmo1^{H/H} Ins2^{Akita/+}* mice. These results suggest that DNA methylation and restoration of IGF-1 signaling mediate at least part of the cardioprotection by B12 in diabetic cardiomyopathy, independently of the direct superoxide scavenging effect of B12. Indeed, an agonist of IGF-1 receptor IGF-1 LONG®R³ (LR3), a methyl donor S-adenosylmethionine (SAMe) and a SOCS1/3 inhibitor pJAK2(1001–1013) partially prevented the changes in cardiac morphology and function in ELMO1 hypermorphic Akita mice (Fig. 7e, f and Suppl. Fig. 6a–j).

## Discussion

In the current work, we demonstrated that a high oral dose of B12 that normalizes the plasma and tissue B12 levels not only protected diabetic mice from developing cardiomyopathy but also effectively reversed early signs of their cardiac dysfunction, independently of blood glucose levels. As a potent superoxide dismutase mimetic, B12 converts superoxide to hydrogen peroxide at a reaction rate approaching SODs[11,12], and it achieved reductions of ROS indicators in diabetic mice to similar levels to those treated with other antioxidants, NAC and tempol. However,

cardioprotection achieved through the reduction of ROS appears partial, as protection by B12 was greater than those achieved by NAC or tempol. Search of additional protective properties led us uncover that B12, enhances IGF-1 signaling, a well-known cardioprotective system[34], which helps complete the explanation of its effectiveness. In contrast, NAC and tempol lack IGF-1 activation property and their functional cardioprotection were partial compared to B12.

Oxidative stress plays a key role in the pathogenesis of diabetic complications. Not surprisingly, therefore, therapeutic attempts with antioxidants have been examined in multiple experimental model systems to prevent diabetes-induced tissue damages. These studies have generally demonstrated beneficial, vascular protective, effects of various antioxidants. However, the outcomes of clinical intervention studies with antioxidants have been elusive[35–38], showing the importance of gaining deeper mechanistic understanding of each agent. To this end, well-defined animal models are crucial. The diabetic model we used in this study is the Akita type 1 diabetic mouse with genetically high levels of *Elmo1* mRNA, at approximately twice normal. Dilated cardiomyopathy with severe systolic and diastolic dysfunction and ultrastructural abnormalities including the dissociation of intercalated discs and mitochondrial fragmentation was uniformly observed in hearts of these mice before complications in other organs, such as in kidneys, begin to manifest[10]. To what extent the high expression of ELMO1 contributes to the B12 effects we observed is currently unknown, yet we observed a similar cardioprotective effect of B12 even in the *Elmo1^{+/+} Ins2^{Akita/+}* mice having normal expression of ELMO1 and less severe cardiomyopathy than *Elmo1^{H/H} Ins2^{Akita/+}* mice.

The presence of low serum B12 levels has been seen in patient with type I diabetes[39]. Patients also have lower plasma IGF-1 levels than non-diabetic individuals[34,40]. In agreement with these observations, we found that the levels of plasma IGF-1, tissue IGF-1 mRNA, and cardiac pIGF1Rβ levels were markedly reduced in *Elmo1^{H/H} Ins2^{Akita/+}* mice as compared with non-

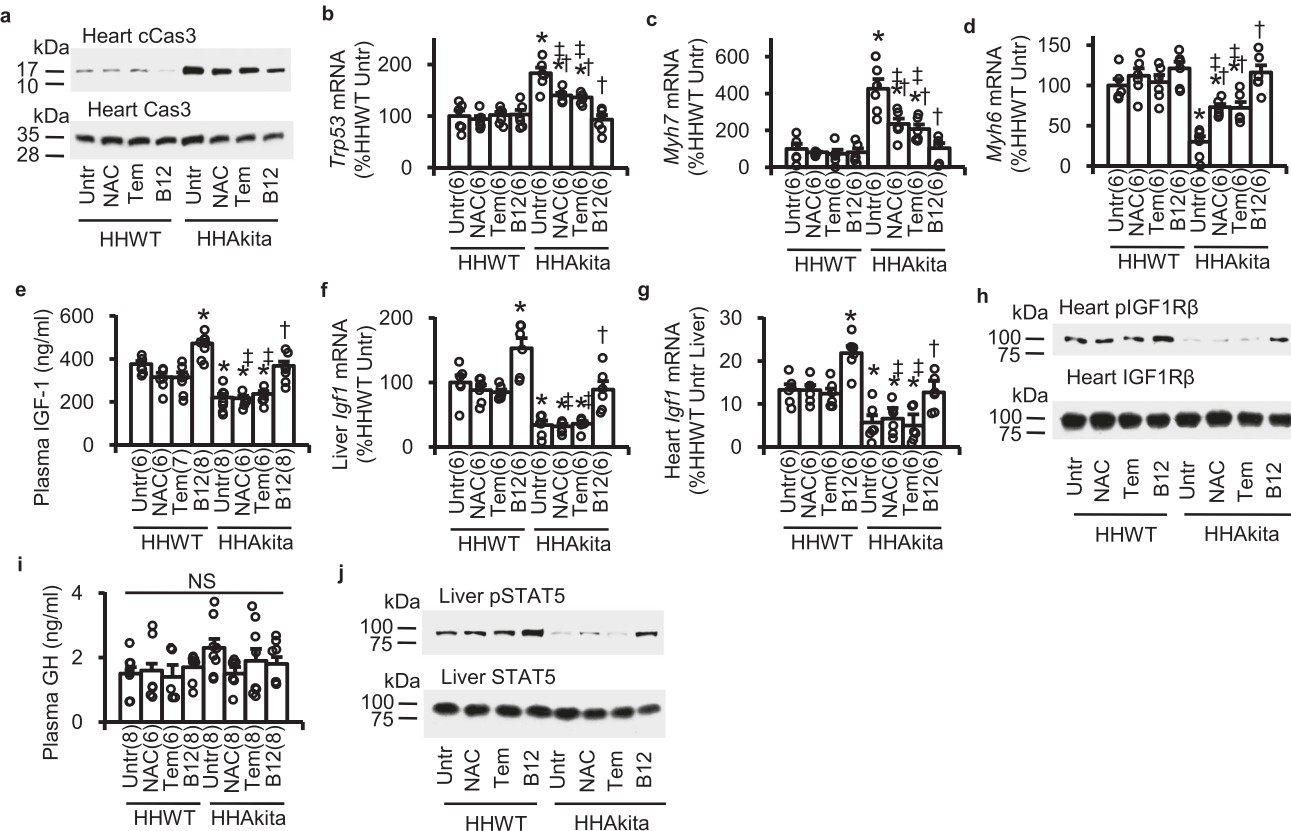

**Fig. 5 Effects of orally administered NAC, tempol (Tem), and B12 on the expression of apoptosis- and cardiomyopathy-associated genes and the parameters related to insulin-like growth factor (IGF)-1 signaling. a** Immunoblots for cardiac cleaved caspase 3 (cCas3) and total caspase 3 (Cas3). **b** Cardiac mRNA levels of the tumor repressor protein 53 gene (*Trp53*). **c** Cardiac mRNA levels of the β-myosin heavy chain gene (*Myh7*). **d** Cardiac mRNA levels of the α-myosin heavy chain gene (*Myh6*). **e** Plasma IGF-1 levels. **f** Hepatic *Igf1* mRNA levels. **g** Cardiac *Igf1* mRNA levels. **h** Immunoblots for cardiac levels of phosphorylated IGF-1 receptor β (pIGF1Rβ) and total IGF1Rβ. **i** Plasma levels of growth hormone (GH). **j** Immunoblots for hepatic phosphorylated pSTAT5 and total STAT5. HHWT, $Elmo1^{H/H}\ Ins2^{+/+}$; HHAkita, $Elmo1^{H/H}\ Ins2^{Akita/+}$. Data are expressed as means ± standard errors. The number of animals analyzed is given in the parentheses in each figure. mRNA levels normalized by β-actin mRNA are expressed relative to the mean levels in unteated HHWT mice as 100%. *$P < 0.05$ vs. untreated (Untr) HHWT mice by Tukey–Kramer Honestly Significant Differences test after one-way ANOVA; †$P < 0.05$ vs. untreated HHAkita mice; ‡$P < 0.05$ vs. B12-treated HHAkita mice; NS, not significant.

diabetic $Elmo1^{H/H}\ Ins2^{+/+}$ mice. Remarkably, these reductions were almost completely restored by the administration of B12. Since neither NAC nor tempol changes plasma and tissue levels of IGF-1, stimulation of IGF-1 system by B12 is unlikely to be due to its antioxidative property. Moreover, since B12 did not change insulin levels, the changes in insulin levels are not the cause of the restoration of IGF-1 receptor signaling by B12. Neither type 1 diabetes nor B12 affected plasma levels of GH, the major regulator of IGH-1 production, in $Elmo1^{H/H}\ Ins2^{Akita/+}$ mice. Since insulin augments the response of IGF-1 mRNA levels to GH in hepatocytes[41], the decreased insulin levels in type 1 diabetes are likely to contribute to the GH resistance[21]. Likewise, insulin deficiency in type 1 diabetes has been found to downregulate IGF-1 receptor signaling in the myocardium[42]. Recently, Roman-Garcia et al.[22] observed that B12 deficiency caused by genetic absence of intrinsic factor decreases hepatic mRNA levels and serum levels of IGF-1 and the phosphorylated IGF-1 receptor β (pIGF1Rβ) levels in hepatocytes and in osteoblasts. The authors suggested that B12 enhances systemic IGF-1 signaling.

The reduced production of IGF-1 in type 1 diabetes likely plays a pathogenic role in diabetic cardiomyopathy and strongly suggests that the cardioprotective effects of B12 is due partly to the restoration of IGF-1 levels. In the current study, we further observed that the co-administration of B12 and linsitinib, an inhibitor of IGF-1 receptor, partially reduced the beneficial effects

of B12 on systolic and diastolic dysfunction in the $Elmo1^{H/H}\ Ins2^{Akita/+}$ mice. Conversely, an IGF-1 receptor agonist, IGF-1 LONG® R[3] (LR3), partially improved the cardiac function in the $Elmo1^{H/H}\ Ins2^{Akita/+}$ mice. Taken together, our results indicate that IGF-1 signaling and superoxide scavenging, both partially but additively mediate the cardioprotective effects of B12 in type 1 diabetes. The mechanisms whereby IGF-1 alleviates diabetic cardiomyopathy suggested by previous studies include activation of the phosphoinositide 3-kinase/Akt pathway, which prevents apoptosis of cardiomyocytes[43], the dissociation of intercalated discs[44], mitochondrial dysfunction[45] and fragmentation[46]. IGF-1 also decreases protein levels of p53 that facilitates cell apoptosis[18]. Consistently, we observed that cleaved caspase 3 levels and the mRNA levels for p53 are lower in the heart of $Elmo1^{H/H}\ Ins2^{Akita/+}$ mice treated with B12 than in those treated with NAC or with tempol. Thus, B12-induced IGF-1 likely prevents apoptosis independently of oxidative stress.

B12 is an essential cofactor for methionine synthase, and liver SAMe levels were also normalized by the B12 administration in diabetic animals and increased in non-diabetic animals. Intriguingly, previous studies and our findings indicated that SAMe[25,26], the substrate of DNMTs, increases the expression of DNMTs. DNMTs have been shown to suppress the expressions of SOCS1 and SOCS3 via methylation of their promoters[31,47]. In parallel to the changes in the SAMe levels in the liver, we found that the

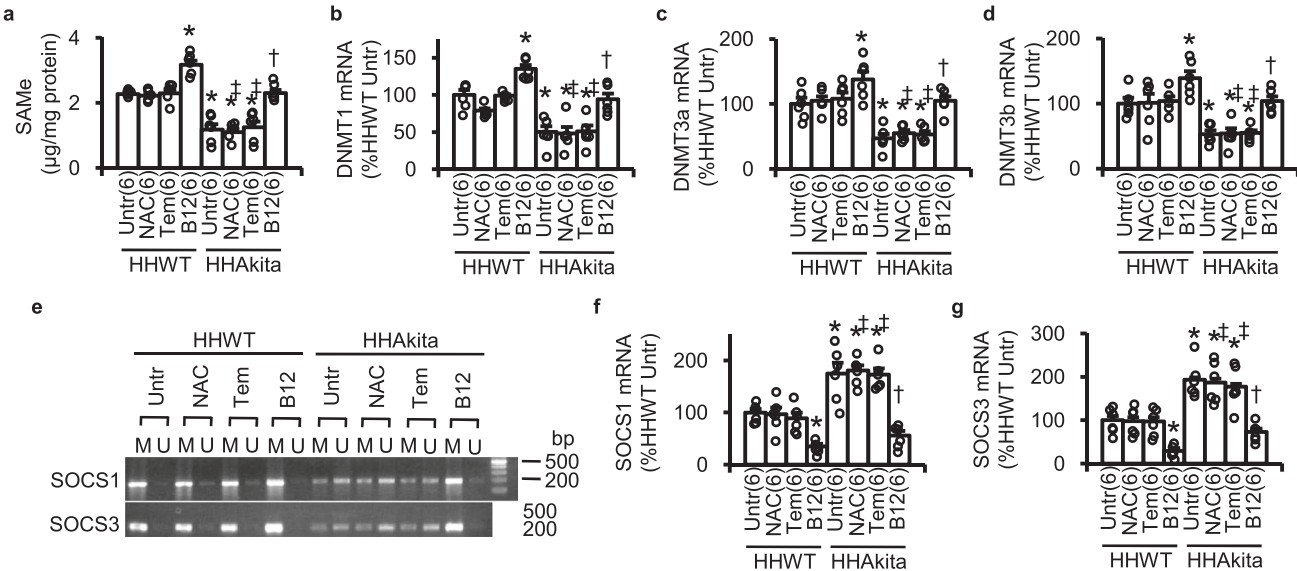

**Fig. 6 Effects of orally administered NAC, tempol (Tem), and B12 on the parameters related to the DNA methylation and expression of the suppressor of cytokine signaling (SOCS) 1 and 3. a** Liver S-adenosylmethionine (SAMe) levels. **b** Liver DNA methyltransferase (DNMT) 1 mRNA levels. **c** Liver DNMT3a mRNA levels. **d** Liver DNMT3b mRNA levels. **e** PCR products amplified with methylation (M) and unmethylation (U)-specific primer sets for the promoters of SOCS1 and SOCS3 using 50 ng bisulfite-treated liver DNA. **f** Liver SOCS1 mRNA levels. **g** Liver SOCS3 mRNA levels. HHWT, $Elmo1^{H/H} Ins2^{+/+}$; HHAkita, $Elmo1^{H/H} Ins2^{Akita/+}$. Data are expressed as means ± standard errors. The numbers of animals analyzed are given in the parentheses. mRNA levels normalized by β-actin mRNA are expressed relative to the mean levels in untreated HHWT mice as 100%. *$P < 0.05$ vs. untreated (Untr) HHWT mice by Tukey–Kramer Honestly Significant Differences test after one-way ANOVA; †$P < 0.05$ vs. untreated HHAkita mice. ‡$P < 0.05$ vs. B12-treated HHAkita mice.

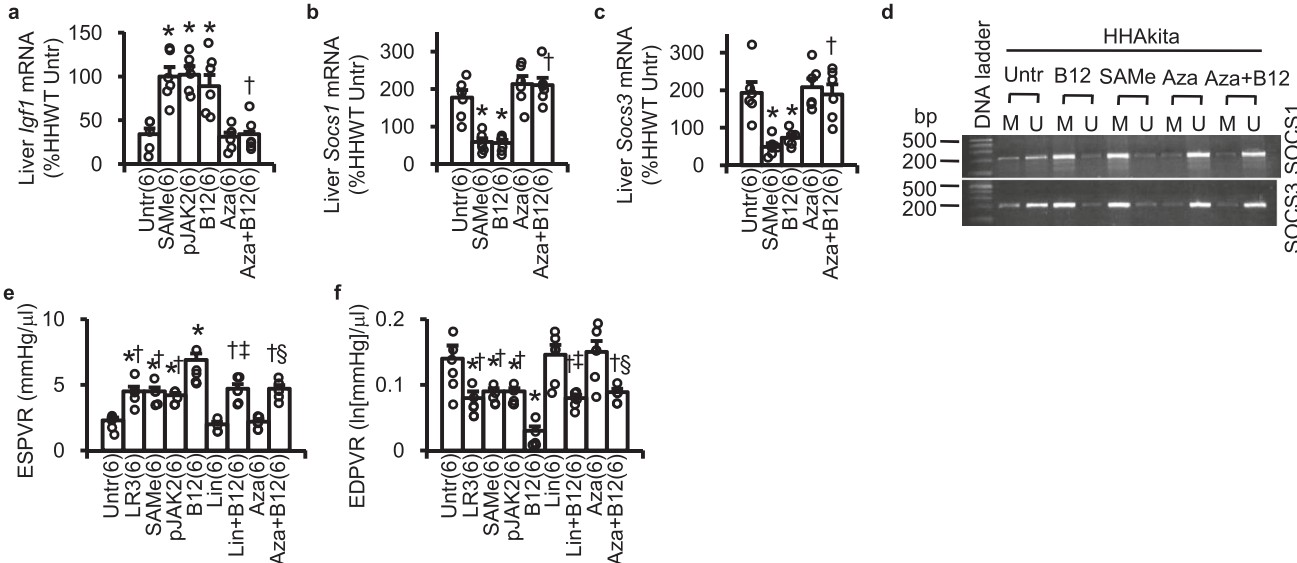

**Fig. 7 DNMTs-SOCS1/3-IGF-1 pathway on the cardioprotective effects of B12. a** Liver *Igf1* mRNA, **b** Liver *Socs1* mRNA, and **c** Liver *Socs3* mRNA of $Elmo1^{H/H} Ins2^{Akita/+}$ mice treated for 16 weeks with S-adenosylmethionine (SAMe; 200 mg/kg/day PO), a SOCS1/3 inhibitor pJAK2(1001–1013) (pJAK2; 50 mg/kg/day IP), B12 alone, and a DNMT inhibitor 5-aza-2′-deoxycytidine (Aza; 10 mg/kg/day IP) with or without B12 compared with those for untreated (Untr) or and treated mice with B12 only (the same as those in Figs. 5 and 6). mRNA levels normalized by β-actin mRNA are expressed relative to the mean levels in HHWT mice as 100%. *$P < 0.05$ vs. Untr; †$P < 0.05$ vs. B12 only. **d** PCR products amplified with methylation (M) and unmethylation (U)-specific primers for the promoters of SOCS1 and SOCS3 using 50 ng of bisulfite-treated liver DNA. The first lane of the panels shows DNA size markers. **e** End-systolic pressure–volume relationship (ESPVR) determined by pressure–volume loop analysis and **f** End-diastolic pressure–volume relationship (EDPVR) of the $Elmo1^{H/H} Ins2^{Akita/+}$ mice assessed after 8 weeks of treatments with a long-acting IGF-1 analog IGF-1 Long R3 (LR3; 1 mg/kg/day IP), SAMe, pJAK2, and an IGF-1 receptor inhibitor linsitinib (Lin; 50 mg/kg/day PO) and Aza with and without B12. Data are expressed as means ± standard errors. *$P < 0.05$ vs. Untr; †$P < 0.05$ vs. B12; ‡$P < 0.05$ vs. Lin; §$P < 0.05$ vs. Aza; by Tukey–Kramer Honestly Significant Differences test.

expression of SOCS1 and SOCS3 were moderated by B12. SOCS1 and SOCS3 have been demonstrated to inhibit the ability of JAK2 to phosphorylate STAT5[33] in response to the GH receptor. Reduced SOCS1/3 in turn facilitate nuclear transfer of phosphorylated STAT5, which in turn stimulates the transcription of

IGF-1 in hepatocytes (Fig. 8). Accordingly, the DNMT-SOCS1/3 cascade is likely to be involved in the mechanism whereby B12 restores the levels of pSTAT5 and IGF-1 mRNA in the liver of the $Elmo1^{H/H} Ins2^{Akita/+}$ mice. Indeed, the DNMT substrate SAMe decreased hepatic SOCS1/3 expression and increased IGF-1

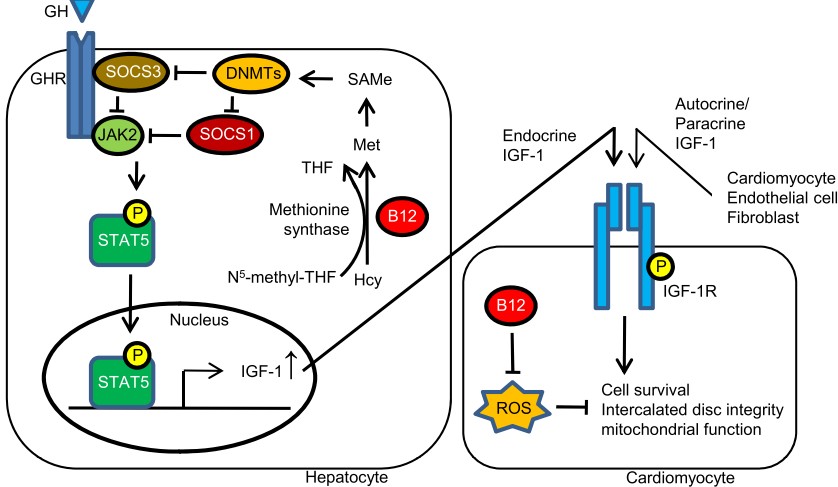

**Fig. 8 A proposed mechanism whereby cobalamin (B12) prevents diabetic cardiomyopathy.** B12 as an essential cofactor of methionine synthase increases hepatic levels of SAMe. SAMe is not only the methyl donor for DNMTs to methylate the promoters of SOCS1 and SOCS3, but also increases the expression of DNMTs, which further enhances the SOCS1/3 methylation. Both SOCS1 and SOCS3 suppress the activity of JAK2, which is the protein responsible for phosphorylating STAT5. The phosphorylated STAT5 translocates into the nucleus and stimulates the transcription of IGF-1 in hepatocytes. Circulating as well as locally produced IGF-1 binds to the cardiac IGF-1 receptor, which is phosphorylated to stimulate the cardioprotective IGF-1 signaling that increases cell survival and ameliorates diastolic and systolic functions in the diabetic heart. Additionally, B12 contributes to the cardioprotection by directly reducing ROS in diabetic cardiomyocytes and producing IGF-1 locally. Met, methionine; Hcy, homocysteine; THF, tetrahydrofolate.

signaling and cardiac function, whereas a DNMT inhibitor 5-aza-2'-deoxycytidine increased the SOCS1/3 expression and decreased IGF-1 signaling and cardiac function. We also note that, while liver is the major source of circulating IGF-1, hearts also produce IGF-1 locally, and we observed that B12 also restored *Igf1* mRNA levels in the heart. Defining the protective roles in diabetic hearts of local paracrine/autocrine functions of IGF-1 produced by cardiomyocytes, cardiac fibroblasts or blood vessels, requires future investigations.

In humans, subclinical B12 deficiency is prevalent in patients with type 1 diabetes[39]. Oral administration of B12 (1 mg/day) together with high doses of B6 and B9 was tested in a large HOPE-2 trial involving 5522 patients with vascular disease (including 2209 diabetics), and the treatment reduced homocysteine levels and the risk of non-fatal stroke but did not benefit diabetic nephropathy. Adverse effects were not significant[48]. In contrast, a small trial (DIVINe study) of a combination therapy of high-dose B vitamins including B12 (1 mg/day) in 238 patients with advanced diabetic nephropathy did not have demonstrable vascular benefit. Instead, the study was terminated early because of adverse observation of decreased glomerular filtration rate and higher cardiovascular events in B-vitamins treated group compared to placebo treated group[49]. Notably, there are multiple important differences in the condition of these human studies and our experiments in mice. First, human patients had already clinically established vascular diseases while most mouse studies employed those at the early stages of diabetic complications. Second, the doses of B12 tested in these human studies were much lower than what we used (equivalent to 50 mg/day intake in humans). Third, the effects of B12 combined with other B vitamins have not been studied in mice. We note that the B12 dose we have given to the diabetic mice did not increase plasma IGF-1 levels beyond the normal levels, while long-term therapeutic use of IGF-1 in human patients has been controversial, because of some side effects[34].

In summary, our finding is notable since the cardiopathic effects in type 1 diabetes are preventable if high doses of B12 are administered orally, even though the diabetic hyperglycemia persists. B12 prevents and reverses the development of diabetic

cardiomyopathy in the ELMO1 hypermorphic Akita mice, at least via direct scavenging of superoxide and the retrieval of IGF-1 signaling via modulating the SAMe-DNMT-SOCS1/3 cascade (Fig. 8). Thus, B12 could be a promising option for preventing and treating cardiomyopathy in patients with diabetes. There is no established upper limit nor toxicity for oral vitamin B12 intake, and it is generally safe according to the Food and Nutrition Board at the Institute of Medicine. In one trial, B12 was given to patients with multiple sclerosis at 60 mg/day without any adverse effects[50]. Nevertheless, any substances have optimal ranges of doses that is subjective to individual health conditions and genetic compositions. Clearly, further studies are necessary to evaluate the safety of long-term use of a high oral dose B12 in oxidative stress reduction and IGF-1 signaling retrieval.

## Methods
**Mice.** Homozygotes for the hypermorphic (H) alleles for the ELMO1 gene (*Elmo1^{H/H}*)[1,2] with heterozygous diabetogenic Akita mutation in the Insulin 2 gene (*Ins2^{Akita/+}*; Jackson Laboratory; Stock No: 003548) were generated by crossbreeding. All experiments used male mice on a C57BL/6J genetic background, because male *Ins2^{Akita/+}* mice develop type 1 diabetes, while female *Ins2^{Akita/+}* mice do not develop diabetes[3]. Unless otherwise stated, vitamin B12 was provided through their drinking water at a dose to achieve an estimated daily B12 intake of 10 mg/kg bw/mouse, starting at 8 weeks of age for 16 weeks, echocardiograms were taken at 8, 16, and 24 weeks, and other biochemical analyses were performed on tissues collected at 24 weeks of age. All mice were kept under husbandry conditions conforming to the National Institutes of Health Guideline for Use and Care of Experimental Animals, and the protocols used were approved by the University of North Carolina Institutional Animal Care and Use Committee. All measurements were taken from distinct samples.

**Dose effects of orally administered B12 on plasma and cardiac levels of B12 and on cardiac function in the diabetic *Elmo1^{H/H} Ins2^{Akita/+}* mice.** Our regular rodent chow (CAT#3002909-203, PicoLab, Fort Worth, Texas) contains ~80 μg/kg of B12, and assuming that wild type C57BL/6 mice at 25 g of body weight eat about 4–5 g of chow and drink 4 ml of water, their average B12 intake was estimated as 0.013 mg/kg bw/day. Our past experience of average food and water consumptions of diabetic *Ins2^{Akita/+}* mice weighing 28 g at 3 to 6 months of age are 8.5 g and 28 ml per day, respectively[51,52]. The estimated B12 from chow averaged 0.024 mg/kg bw/day. *Ins2^{Akita/+}* mice typically become maximally hyperglycemic by age 8 weeks[53], and our previous work[10] demonstrated that *Elmo1^{H/H} Ins2^{Akita/+}* mice develop severe cardiomyopathy by age 16 weeks. We, therefore provided 8-week-old *Elmo1^{H/H} Ins2^{Akita/+}* mice with B12 in their drinking water calculated to be

averaged daily intake at 1, 10, and 100 mg/kg bw/day of supplementation for 8 weeks. Food and water consumption between groups were not grossly differed. No obvious side effects were observed by oral administration of the high doses of B12. At 16 weeks of age, echocardiography was carried out and mice were sacrificed to measure plasma and cardiac concentration of B12 (Mouse Vitamin B12 ELISA kit; MyBioSource).

**Preventive and therapeutic effects of B12 on the diabetic cardiomyopathy in Elmo1$^{H/H}$ Ins2$^{Akita/+}$ mice.** Mice were separated into three groups at age 8 weeks: the control group without receiving B12 ($N = 10$), the group receiving B12 in drinking water at 10 mg/kg bw/day intake for 16 weeks starting at age 8 weeks (Preventive group; $N = 9$), and the group receiving B12 in drinking water at 10 mg/kg bw/day for 8 weeks starting at age 16 weeks (Therapeutic group; $N = 10$). Echocardiography was performed in three groups at age 8, 16, and 24 weeks. The heart functions were measured at 8, 16, and 24 weeks of age. Mice were sacrificed at age 24 weeks and tissues were collected to study histology, gene expression, the plasma and cardiac parameters of oxidative stress and mitochondrial function.

**Comparing the preventive effects of B12 and other antioxidative agents on diabetic cardiomyopathy.** Two additional groups of mice were studied: the Elmo1$^{H/H}$ Ins2$^{Akita/+}$ mice ($N = 9$) receiving NAC in their drinking water (Millipore Sigma; 1000 mg/kg bw/day) for 16 weeks starting at age 8 weeks, and the Elmo1$^{H/H}$ Ins2$^{Akita/+}$ ($N = 8$) receiving tempol in drinking water (Santa Cruz Biotechnology; 200 mg/kg bw/day) for 16 weeks starting at age 8 weeks, Echocardiograms were taken at 16 and 24 weeks of age and mice were sacrificed at 24 weeks of age for histological and biochemical analyses. The same treatment protocols including with B12 at 10 mg/kg bw/day were applied to the non-diabetic Elmo1$^{H/H}$ Ins2$^{+/+}$ mice ($N = 8$–9 in each group).

**Roles of IGF-1 and DNA methylation in the beneficial effects of B12 in diabetic cardiomyopathy.** Seven additional groups of Elmo1$^{H/H}$ Ins2$^{Akita/+}$ mice ($N = 8$ in each group) were treated with various agents for 16 weeks starting at age 8 weeks; IGF-1 LONG® R$^3$ intraperitoneally (a long-acting IGF-1 analog; Sigma; 0.3 mg/kg bw/day), linsitinib by gavage (an IGF-1 receptor inhibitor; AdooQ Bioscience; 50 mg/kg/day) with or without B12 in drinking water (10 mg/kg bw/day), S-adenosylmethionine in drinking water (a methyl donor for DNA methylation; Swanson; 200 mg/kg bw/day), tyrosine1007-phosphorylated JAK2(1001–1013) intraperitoneally (pJAK2; a SOCS1/3 inhibitor; Psyclo Peptide, Inc.; 10 mg/kg bw/day), and 5-aza-2'-deoxycytidine intraperitoneally (a DNA methyltransferase inhibitor; Cayman Chemical; 20 mg/kg bw/day) with or without B12 in drinking water (10 mg/kg bw/day).

All mice were kept under husbandry conditions conforming to the National Institutes of Health Guideline for Use and Care of Experimental Animals as approved by the Institutional Animal Care and Use Committee.

**Echocardiography.** Heart functions of conscious mice were analyzed by the Vevo 2100 ultrasonograph system (FUJIFILM VisualSonics) with a 30-MHz transducer in the Institutional Cardiovascular Physiology and Phenotyping Core according to the American Society of Echocardiography guidelines. All measurements were performed by an investigator who is highly experienced with the system and the data were analyzed with Vevo 2100 Workstation 1.6.0. software. The systolic blood pressure was measured with a tail-cuff method. The heart rate was echocardiographically determined. Since measurements of different groups of live mice were done at different times, we measured smaller numbers (2–4) of HHWT and untreated HHAkita mice at least every 6 months to ensure the consistency of the ultrasonography and tail-cuff systems.

**Pressure–volume (PV) loop analysis.** The mice are anesthetized, intubated, and placed on a ventilator. A ventral midline skin incision is then made and the thoracic cavity is entered through the sternum. A 1.2 Fr admittance PV catheter (Scisense) is introduced into the left ventricle using a 20-gauge needle. After instrumentation is established and PV measurements are obtained, the inferior vena cava is briefly occluded to obtain alterations in venous return for determination of end-systolic and end-diastolic pressure relations. Measurements were done in the Institutional Cardiovascular Physiology and Phenotyping Core.

**Histology and Immunofluorescence.** To obtain tissue samples for histology, the left ventricle was punctured with a 23-gauge needle and perfused with PBS for 3 min and with 4% paraformaldehyde for 5 min, after cutting the inferior vena cava. Thereafter the tissues were dissected out and put in 4% paraformaldehyde for at least 3 days. They were then paraffin embedded and sectioned. Sections were prepared by the Center for Gastrointestinal Biology and Diseases Histology Core and imaged on an Olympus BX61 microscope. Proportions of fibrotic areas were quantified in cross-sections of the heart with Heidenhain's AZAN trichrome staining. Alexa 647-conjugated wheat germ agglutinin, Goat anti-α-actinin antibody (ab190872, Abcam), Alexa 488-conjugated goat anti-rabbit IgG antibody (Thermo Fisher Scientific), and DAPI (Thermo Fisher Scientific) were used to study the cross-sectional areas of cardiomyocytes in heart sections. For electron microscopy, grids were prepared by the Institutional Microscopy Services

Laboratory and imaged on a Zeiss TEM 910 transmission electron microscope. The volume density of mitochondria was determined by the probability of mitochondrial presence at the crossings of the grid on electron photomicrographs[54]. Morphology of mitochondria were scored from at least 30 mitochondria in each animal in a blinded manner ($N = 6$ in each group).

**Biological parameters.** Following biochemical assays were carried out together using tissues kept frozen at −80 ºC. Cardiac citrate synthase activity was studied with Citrate Synthase Activity Colorimetric Assay Kit (BioVision). Cardiac production of H$_2$O$_2$ was studied with Amplex® Red Hydrogen Peroxide/Peroxidase Assay Kit (Thermo Fisher Scientific). Tissue levels of reduced glutathione (GSH) and oxidized glutathione (GSSG) were studied with GSH/GSSG Assay kit (Millipore Sigma). The levels of B12 in the plasma and heart lysate, 4-HNE levels in the heart lysate, S-adenosylmethionine (SAMe) in the liver lysate, plasma growth hormone were studied with ELISA kits (Vitamin B12 ELISA kit, MyBiosource, Inc.; Lipid Peroxidation (4-HNE) Assay kit, Abcam; SAMe ELISA kit, Cell Biolabs Inc.; Mouse Growth Hormone ELISA kit, Crystal Chem).

**Western blotting.** The heart was homogenized in radioimmunoprecipitation assay (RIPA) buffer (50 mM Tris-HCl, pH 7.4, 150 mM NaCl, 50 mM β-glycerophosphate, 30 mM NaF, 2 mM EDTA, 2 mM EGTA, 30 mM Na$_4$P$_2$O$_7$, 2 mM Na$_3$VO$_4$, 1% Triton X-100) containing protease inhibitors (cOmplete$^{TM}$, Roche). The homogenate was centrifuged at 1000 g to pellet the nucleus and debris. The heat-dissociated proteins (40 μg/lane) were fractionated by SDS-PAGE (Mini-PRO-TEAN® TGX, Bio-Rad) and detected with Western blot by chemiluminescence (SuperSignal$^{TM}$ West Pico PLUS, Thermo Fisher Scientific). Blots developed on the chemiluminescence film (Amersham Hyperfilm ECL; GE Healthcare) were quantified using ImageJ software (NIH). The antibodies used for Western blot are as follows:Y1135-phosphorylated IGF-1 receptor β (pIGF1Rβ; Rabbit mAb; DA7A8; Cell Signaling Technology), IGF-1 receptor-β (IGF1Rβ; Rabbit pAb; 3027; Cell Signaling Technology), Y694-phosphorylated signal transducer and activator of transcription 5 (pSTAT5; Rabbit mAb; C11C5; Cell Signaling Technology), total STAT5 (Rabbit mAb; D206Y; Cell Signaling Technology), cleaved caspase 3 (Rabbit mAb; 9664; Cell Signaling Technology), total caspase 3 (Rabbit mAb; 8610; Cell Signaling Technology), and β-actin (HRP-conjugated Rabbit mAb; 3683; Cell Signaling Technology).

**Mitochondrial complex activity.** The activity of cardiac mitochondrial complexes I, IV and V was studied with Abcam microplate assay kits (Complex I Enzyme Activity Assay Kit, ab109721; MitoTox$^{TM}$ Complex IV OXPHOS Activity Assay Kit ab109906; MitoTox$^{TM}$ Complex V OXPHOS Activity Assay Kit ab109907).

**Quantitative reverse transcription-PCR.** Total RNA was extracted from different tissues and the mRNAs were assayed by quantitative reverse transcription-PCR. The primers and the probes used to measure the mRNA levels are shown in Supplemental Table 1.

**Methylation-specific PCR (MSP) analysis.** Fifty nanogram of bisulfite-treated hepatic genomic DNA (EpiTect® Fast Bisulfite Kit; Qiagen) was amplified (PrimeSTAR® HS DNA polymerase with GC buffer; Takara) by using methylation- or unmethylation-specific primer sets for SOCS1 and SOCS3. The primer sequences used to amplify the methylated SOCS1 gene were 5'-CGAGGAATTAGGTCGG GAGC-3' (forward) and 5'-TCGACCCTTCTTAAAACCCG-3' (reverse), and the primer sequences used to amplify the unmethylated SOCS1 gene were 5'-TTGA GGAATTAGGTTGGGAGT-3' (forward) and 5'-CTCAACCCTTCTTAAAACC CA-3' (reverse). The primer sequences used to amplify the methylated SOCS3 gene were 5'-AGGGGTCGTTGTTAGGAAC-3' (forward) and 5'-TTCCTAAAACTA CCCGACCG-3' (reverse), and the primer sequences used to amplify the unmethylated SOCS3 gene were 5'-GGAGGGGTTGTTGTTAGGAAT-3' (forward) and 5'-GATTCCTAAAACTACCCAACCA-3' (reverse). MSP analysis was performed in a thermal cycler with the following conditions: 95 °C for 12 min, 40 cycles of 95 °C for 20 s, 58 °C for 20 s and 72 °C for 30 s, and a final extension of 10 min at 72 °C.

**Statistics and reproducibility.** All measurements were taken from biologically distinct samples. Data are expressed as means ± standard errors. To compare groups, we used one-factor or two-factor analysis of variance (ANOVA). Post hoc pairwise comparisons were performed by Tukey–Kramer Honestly Significant Differences test (JMP 13; SAS Institute Inc.). Reproducibility including biologically independent sample sizes is stated in each figure legend.

**Reporting summary.** Further information on research design is available in the Nature Research Reporting Summary linked to this article.

## Data availability
The data that support the findings of this study are available from the corresponding author upon reasonable request.

ARTICLE

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

## Acknowledgements

This work was supported by an NIH Grant R01HL049277. We thank Ms. Carolyn B. Suitt prepared heart sections for immunofluorescence, Ms. Kristen K. White, Ms. Victoria J. Madden, and Dr. Pablo Ariel for assistance in light and transmission electron microscopy. We also thank Dr. Brian C. Cooley for performing the pressure–volume loop studies.

## Author contributions

M.K., O.S. conceived the project and designed experiments. M.K., P.V.R., J.R.H., R.G., and J.C.W. performed experiments. J.M.T., J.C.J., O.S., and N.M. helped project conceptualization and data interpretation. M.K. and N.M. wrote the manuscript with input of other authors.

## Competing interests

The authors declare no competing interests.
