## [Peer Review File · Communications Biology]

Reviewers' comments:

Reviewer #1 (Remarks to the Author):

In the present study authors have investigated the protective effect of B12 in mice having high expression levels of ELMO1 without and with Akita diabetogenic mutation and concluded that oral administration of B12 for 16 weeks prevented diabetic cardiomyopathy by superoxide scavenging property as well as retrieving insulin like growth factor, IGF-1, signaling via modulating the DNMT-SOCS1/3 cascade. The paper is interesting; however, I have some comments which need to be addressed

Major comments

1. Only a single knockout model of diabetic cardiomyopathy is used to prove the therapeutic potential of B12. The study will benefit greatly if other models of diabetic cardiomyopathy specifically diet/lifestyle induced will be used to prove the beneficial effect which will be much more relevant with today's rapid increase in cardiovascular complications due to lifestyle factors.
2. Although authors have thoroughly measured other parameters (FBG, Insulin etc), What was the effect of B12 administration on other biochemical markers, triglycerides, FFA, TNF, TGF Beta etc
3. In Figure 1 F: Histological staining by WGA lectin of the heart revealing cross-sectional areas of cardiomyocytes, looking at the images, not much difference is found between B12, tempol and NAC.
4. Diabetic cardiomyopathy is a multifactorial disease with number of pathways involved. The main thing is there is a consistent increase in blood glucose which leads to glucotoxicity which leads to formation of advanced glycation end products and oxidative stress. However, based on table 2, B12 is having no effect on fasting blood glucose. How about the effect on GTT? Suitable justification is required.
5. In the study high doses of Cyanocobalamin have been used. Additional experiments need to be performed invitro or invivo to prove toxicity is within acceptable range.
6. The efficacy of B12 in comparison to tempol and NAC is statistically not shown based on the figures presented. For better understanding, to prove therapeutic efficacy of B12, this should be included.

Reviewer #2 (Remarks to the Author):

In this manuscript, the authors examined the impact of B12 on cardiac function in a diabetic setting. They also attempted to dissect the molecular mechanism whereby B12 evokes its effects. Using a type 1 diabetes mouse model that overexpresses Elmo1, the authors demonstrated that this model develops diabetic cardiomyopathy with severe contractile dysfunction which was prevented and or reversed with B12 supplementation. Mechanistically, the authors demonstrated that B12 mediated its effects through activation of IGF-1 signaling via modulation of the DNMT-SOCS1/3 pathway. The study is well done, with a number of controls included. However, it is generally incremental in nature and didn't provide new findings since the effect of B12 and IGF-1 on cardiac function has widely been reported in the literature.

- 1- The effects of B12 on the heart appear to be indirect, liver-mediated. This raises the possibility that other organs might as well contribute the overall benefits of B12 on the heart. The authors are strongly encouraged to perform in vitro studies using isolated adult cardiomyocytes to demonstrate a direct effect of B12 on myocyte physiology.
- 2- The persistence of hyperglycemia in the model is a concern and its effect on cardiac performance can not be ruled out. Please address this point.
- 3- It is very well known that acute supplementation of insulin are beneficial to the heart; however, chronic and long term use of insulin are detrimental. How can the authors demonstrate that chronic activation of IGF-1 is safe and beneficial and not the opposite? This needs to be addressed and confirmed.

4- It has been reported in the literature that the Akita heart is atrophic. Have the authors considered looking into this aspect and how B12 affects atrophy?

5- I believe the major concern I have apropos this study is the data presented regarding the development of hypertrophy following B12 supplementation. In Fig.1F, B12 appears to promote enlargement of cardiomyocyte size, a sign of hypertrophy. This was paralleled by an increase in LVPWd wall thickness (Suppl Fig 1M) and a decrease in LV internal cavity (Suppl Fig 1N). These are features of a concentric hypertrophy phenotype. So, does B12 prevents heart failure but promotes hypertrophy? Please address . Also, the authors show in Fig. 1F enlargement of cardiomyocyte cross sections, while in Fig 3C they showed B12 decreased expression of MyH7, a marker of hypertrophy. On one hand, B12 increases myocyte size and the other hand reduces markers of enlarged myocytes. This is contradictory and needs to be addressed as well.

Response to the Reviewer 1

1. Only a single knockout model of diabetic cardiomyopathy is used to prove the therapeutic potential of B12. The study will benefit greatly if other models of diabetic cardiomyopathy specifically diet/lifestyle induced will be used to prove the beneficial effect which will be much more relevant with today's rapid increase in cardiovascular complications due to lifestyle factors.

Our model of enhanced diabetic cardiomyopathy is a type 1 model with increased oxidative stress, because it carries genetic variants that increase ELMO1 gene expression, mimicking the human variant that has been established to enhance nephropathy in patients with type 1 diabetes. As the reviewer correctly points out, type 2 diabetes is rapidly increasing in human population and has increased cardiovascular complications. However, the etiology of type 2 diabetes is complex. Additionally, cardiovascular consequences of patients are more complicated than in type 1 diabetes because of the high frequency of co-morbidities including obesity, hypertension and atherosclerosis. It has been reported that vitamin B12 deficiency is not associated with patients with type 2 diabetes, unless treated with biguanides (Berger, W. Incidence of severe sideeffects during therapy with sulfonylureas and biguanides. *Hormone and metabolic research* 15, 111-115, 1985). Production of IGF-1 in response to growth hormone is initially increased in patients with Type 2 diabetes (Grecu, E.O., Spencer, E.M., White, V.A. & Sheikholislam, B.M. Exaggerated somatomedin-C response to human growth hormone infusion in patients with type II diabetes mellitus. *The American journal of the medical sciences* 287, 7-10, 1984) after which it gradually decreases with increasing age (Janssen, J.A. & Lamberts, S.W. The role of IGF-I in the development of cardiovascular disease in type 2 diabetes mellitus: is prevention possible? *European journal of endocrinology* 146, 467-477, 2002). We totally agree with the reviewer that it is very important to examine effects of B12 in type 2 diabetes, and we intend to examine the effects of ELMO1 and of B12 in our future studies, but it is beyond the scope of the current work. We have changed the title into "Cyanocobalamin prevents cardiomyopathy in type 1 diabetes by modulating oxidative stress and IGF-1 signaling" to reflect that our study used a type 1 model.

2. Although authors have thoroughly measured other parameters (FBG, Insulin etc), What was the effect of B12 administration on other biochemical markers, triglycerides, FFA, TNF, TGF Beta etc.

In response to the reviewer's comment, we measured plasma triglyceride levels (in the supplemental Table 1) and cardiac mRNA level of the Tgfb1 gene (Supplemental Figure 4B) and described in the text on page 5 and 9 respectively.

3. In Figure 1 F: Histological staining by WGA lectin of the heart revealing cross-sectional areas of cardiomyocytes, looking at the images, not much difference is found between B12, tempol and NAC.

It may be difficult to appreciate the differences from histological staining alone, but when cross-sectional areas were quantitated from multiple sections in each mouse, cardiomyocytes in mice given B12 were significantly larger than those given NAC or tempol in diabetic mice

(Figure 3H), but they did not differ from those in the non-diabetic mice. Thus, the cell sizes in B12 treated mice were normal and not hypertrophic. The significant differences between B12 treatment and NAC/tempol treatment are indicated by “‡” in figures.

4. Diabetic cardiomyopathy is a multifactorial disease with number of pathways involved. The main thing is there is a consistent increase in blood glucose which leads to glucotoxicity which leads to formation of advanced glycation end products and oxidative stress. However, based on table 2, B12 is having no effect on fasting blood glucose. How about the effect on GTT? Suitable justification is required.

As the reviewer points out, diabetic cardiomyopathy is a multifactorial disease with a number of pathways involved. Our finding is that B12, NAC, and tempol mitigated diabetic cardiomyopathy without significant changes in blood glucose, suggesting that the restoration of cardiac function by antioxidants in our model is not dependent on blood glucose. Instead, we suggest that the suppression of ROS and enhancement of IGF-1 production caused by B12 are the mechanisms whereby B12 alleviated diabetic cardiomyopathy.

5. In the study high doses of Cyanocobalamin have been used. Additional experiments need to be performed in vitro or in vivo to prove toxicity is within acceptable range.

Toxicity is an important point. The high dose we used in our study normalizes plasma and tissue levels of B12 in the diabetic mice, but they do not exceed the levels of non-diabetic mice as indicated in the first section of the results. There is no known toxicity of B12 described. However, we agree with the reviewer that long-term supplementation with high dose B12 has not been established. GTT in Akita mice with very high glucose levels are not easy to interpret, but will try in our future studies. We have discussed the potential toxicity issue in the discussion in line 7 on page 18.

6. The efficacy of B12 in comparison to tempol and NAC is statistically not shown based on the figures presented. For better understanding, to prove therapeutic efficacy of B12, this should be included.

The statistical differences between B12 and the other antioxidants were shown with the symbol “‡” in figures.

Thank you very much for your insightful comments. We believe that our revised manuscript has improved greatly by addressing these comments.

Response to the Reviewer 2.

The study is well done, with a number of controls included. However, it is generally incremental in nature and didn't provide new findings since the effect of B12 and IGF-1 on cardiac function has widely been reported in the literature.

The reviewer states that the work is “incremental in nature and didn't provide new findings”. It is true that both B12 and IGF-1 has been studied for a long time. And we agree that cardioprotective role of IGF-1 is now well established, including in diabetic cardiomyopathy. B12 is less so. Most of the studies speculate/predict based on the association of B12 deficiency and diabetes, and this is the first report experimentally demonstrating its therapeutic use as a single oral agent to mitigate diabetic cardiomyopathy and providing its mechanisms in an animal model. Role of B12 in restoring IGF-1 to mitigate development of diabetic cardiomyopathy is a new mechanistic finding.

1- The effects of B12 on the heart appear to be indirect, liver-mediated.

Yes and No. We showed that IGF-1 mRNA levels are increased in both the liver and the heart (Figure 5F and 5G), suggesting that IGF-1 is acting in both endocrine and autocrine/paracrine manners. We clarified this point in the result section from line 15 on page 10 as “We observed that plasma concentrations of IGF-1 and both hepatic and cardiac levels of IGF-1 mRNA were markedly lower in untreated diabetic mice than non-diabetic mice (Figure 5E-G)”, and in the discussion section in line 11 on page 14 as “Indeed, B12 is capable of inducing hepatic and cardiac expression of IGF-1 and increasing circulating IGF-1 independent of diabetes, suggesting that IGF-1 is acting in both endocrine and autocrine/paracrine manners.”

Local paracrine/autocrine production of IGF-1 could be derived from cardiomyocytes, cardiac fibroblasts and/or blood vessels. The nature and extent that local IGF-1 contributes to the protection from cardiomyopathy in the diabetic animals therefore requires further investigation (as we described in the discussion in line 15 on page 14). Although primary cardiomyocytes can be isolated and cultured, diabetic condition cannot be fully reproduced in primary cells in culture of cardiomyocytes.

2- The persistence of hyperglycemia in the model is a concern and its effect on cardiac performance cannot be ruled out. Please address this point.

Our finding is that even though the hyperglycemia persists after B12 is administered, the cardiac functions were effectively restored by B12 in our model of Type 1 diabetes. Accordingly, the mechanisms underlying the diabetic cardiomyopathy could largely be independent of plasma glucose levels. Note, however, the hyperglycemia in these mice are very high and probably reached the maximal levels. We have modified this point in the discussion by describing, “In the current work, we demonstrated that a high oral dose of B12 that normalizes the plasma and tissue B12 levels not only protected diabetic mice from developing cardiomyopathy but also effectively reversed early signs of their cardiac dysfunction, independently of blood glucose levels.”

3- It is very well known that acute supplementation of insulin are beneficial to the heart; however, chronic and long term use of insulin are detrimental. How can the authors demonstrate that chronic activation of IGF-1 is safe and beneficial and not the opposite? This needs to be addressed and confirmed.

IGF-1 recombinant proteins are approved by FDA and EMEA for treatment of severe primary IGF-1 deficiency (Laron's Syndrome), although IGF-1 has some side effects in clinical use (Fintini, D., Brufani, C. & Cappa, M. Profile of mecasermin for the long-term treatment of growth failure in children and adolescents with severe primary IGF-1 deficiency. *Therapeutics and clinical risk management* 5, 553-559, 2009). In the discussion, we have addressed this point in line 15 on page 17, "Clearly, further studies are necessary to evaluate the safety of long-term use of a high oral dose B12 in oxidative stress reduction and IGF-1 signaling retrieval." In addition, we have stressed that in our work "the B12 dose of 10 mg/kgBW/d normalized its levels in plasma and in hearts in the diabetic mice" (Figure 1A and 1B) in the results inline 21 on page 5. Likewise, we have made this point clearer in the discussion as "the B12 dose of 10 mg/kgBW/d in diabetic mice also normalized their plasma IGF-1 levels, but did not over increase further", in line 12 on page 14. However, the same dose of B12 increased plasma IGF-1 by about 20% in non-diabetic controls, and B12 at 10X higher dose of 100 mg/kgBW/d increased the plasma IGF-1 levels to about 140% normal in diabetic mice. "Although we did not detect any adverse effects in mice given and no-upper limit of oral administration of B12 has been determined and regarded generally safe, as any agents we need to be careful not to overdose for longer time usage." We made this point clearer in the discussion in line 9 on page 18 of the revised manuscript.

4- It has been reported in the literature that the Akita heart is atrophic. Have the authors considered looking into this aspect and how B12 affects atrophy?

Yes. We also found that the cross-sectional area of cardiomyocytes was decreased in Akita diabetic mice (Figure 3H) and restored by B12.

5- I believe the major concern I have apropos this study is the data presented regarding the development of hypertrophy following B12 supplementation. In Fig.1F, B12 appears to promote enlargement of cardiomyocyte size, a sign of hypertrophy. This was paralleled by an increase in LVPWd wall thickness (Suppl Fig 1M) and a decrease in LV internal cavity (Suppl Fig 1N). These are features of a concentric hypertrophy phenotype. So, does B12 prevents heart failure but promotes hypertrophy? Please address.

The reviewer's comment made us realize that our descriptions and the use of words "increase/decrease" were confusing. In describing Figure 3 in line 3 on page 9, we have clarified that B12 restored/normalized the levels to those of non-diabetic mice, which are illustrated by the dotted lines in Figure 3, or given as columns of untreated HHWT mice and described that "Histological staining by wheat germ agglutinin (WGA) lectin of the heart revealed that the cross-sectional areas of cardiomyocytes were significantly smaller in untreated diabetic mice compared to non-diabetic mice (Figure 3F, 3H). B12 prevented these histological changes almost completely in the *Elmo1^{H/H} Ins2^{Akita/+}* mice, while prevention by NAC and tempol were partial." The description of the "increase/decrease" were relative to

untreated diabetic mice, which were either decreased or increased compare to untreated non-diabetic mice. In the revised manuscript, we have avoided the use of increase/decrease throughout the text.

In regard to the concentric hypertrophy, the LVPWd and LVIDd in mice given B12 at a dose of 10 mg/kgBW/d were not significantly different from those in nondiabetic mice (dotted line). The LVPWd was larger in mice treated at a dose of 100 mg/kgBW/d was larger by about 30% than non-diabetic mice, with normal LVIDd. At present, there is no convincing evidence for concentric hypertrophy in B12 treated ELMO1 hypermorphic Akita diabetic mice in the current study.

Also, the authors show in Fig. 1F enlargement of cardiomyocyte cross sections, while in Fig 3C they showed B12 decreased expression of Myh7, a marker of hypertrophy. On one hand, B12 increases myocyte size and the other hand reduces markers of enlarged myocytes. This is contradictory and needs to be addressed as well.

As regard to the greater myocyte size and reduced expression of Myh7, they are so compared to untreated diabetic mice. But the levels were both normalized compared to non-diabetic mice. We have made the normalizations clearer by avoiding the use of increase/decrease and by defining each comparison in the text. Normalization of Myh7 levels is another sign that B12 treated hearts are not hypertrophic. Additionally, it was previously reported that the expression of Myh7 in cardiomyocytes is not associated with the bigger cell size, but with fibrosis (Pandya, K., Kim, H.S. & Smithies, O. Fibrosis, not cell size, delineates beta-myosin heavy chain re-expression during cardiac hypertrophy and normal aging in vivo. *Proceedings of the National Academy of Sciences of the United States of America* **103**, 16864-16869, 2006).

Thank you very much for your insightful comments. They were very helpful and let us realize that the previous descriptions were occasionally confusing, and we have edited them and clarified each point addressed in the revised manuscript. We believe our manuscript has improved greatly.

REVIEWERS' COMMENTS:

Reviewer #1 (Remarks to the Author):

The authors have addressed the questions raised by the reviewer. It can be accepted for publication.

Reviewer #2 (Remarks to the Author):

The authors have addressed my previous comments.